# Online Methodology for Separating the Power Consumption of Lighting Sockets and Air-Conditioning in Public Buildings Based on an Outdoor Temperature Partition Model and Historical Energy Consumption Data

**Tianyi Zhao** [iD], **Chengyu Zhang** [iD], **Terigele Ujeed and Liangdong Ma** *

Institute of Building Energy, Dalian University of Technology, Ling Gong Rd, Dalian 116024, China; zhaotianyi@dlut.edu.cn (T.Z.); jlhdzcy@mail.dlut.edu.cn (C.Z.); Terigele_DUT@126.com (T.U.)
* Correspondence: liangdma@dlut.edu.cn; Tel.: +86-0411-84707753

**Abstract:** Among sub-items of energy consumption in public buildings, lighting sockets play an important role in energy-saving analysis. So, the energy consumption data quality of lighting sockets is important. However, limited by the initial cost of energy monitoring platform, it is difficult to install electricity meters covering all branches and to retrofit the incompact classification electricity branches, which results in a mixture of the lighting socket energy consumption and other components. In this study, a separation methodology is proposed. First, the abnormal data in the energy monitoring platform are cleaned and screened using a clustering algorithm. Second, the average outdoor air temperature partitioning model (OATPM) method and the k-nearest neighbor (KNN) clustering algorithm method are proposed for identifying and separating the abnormal data. These two methods have complementary advantages in the best applicable scenarios, including calculation accuracy and other aspects. The verification results for three buildings show that the relative error of this separation methodology is less than 15%. Finally, this paper presents the optimization parameters of the KNN method. Through this methodology, building managers need only historical data in an energy monitoring platform to separate the combined power consumption of the lighting sockets and air-conditioning online, independent of detailed information statistics.

**Keywords:** building energy monitoring platform; lighting socket power consumption; separation of energy consumption data; k-nearest neighbor clustering algorithm; average outdoor air temperature partitioning model





## 1. Introduction

With increasing urbanization, research in building energy efficiency is becoming more important on a global basis. In 2018, total building energy consumption was 899 million tons of standard coal equivalent (tce), accounting for 20.62% of the total energy consumption in China. In addition, public building energy consumption was 346 million tce, accounting for 38.53% of the total building energy consumption [1,2]. Thus, energy efficiency in public buildings has become a necessity to study the energy-saving potential of different sub-items and energy efficiency strategies. Scholars from various countries study the electricity consumption characteristics of public buildings with different functions. Lee et al. found that the ratio of electricity consumption for air-conditioning, office equipment, elevators, and lighting sockets in commercial buildings was 43%, 17%, 7%, and 34%, respectively, according to an investigation of 16,000 commercial buildings in Hong Kong [3–5]. These results indicate that air-conditioning and lighting sockets in public buildings play an important role in building energy consumption, of which operation characteristics are worthy of special attention. Benavente-Peces et al. used various machine learning technique classifiers to analyze and classify building energy efficiency, and they

demonstrated that reliable classification is feasible with a few featured parameters [6]. Suzane A. Monteiro et al. developed a methodology of energy efficiency for lighting and air-conditioning systems in buildings using a multi-objective optimization algorithm [7].

In recent years, a large number of studies have focused on the study of human behavior models, because occupant behavior is complex and requires an interdisciplinary approach to be fully understood. On the one hand, occupant behavior is influenced by external factors such as culture, economy, and climate, as well as internal factors such as individual comfort preference, physiology, and psychology. On the other hand, the occupants' interactions with building systems strongly influence building operations and thus energy use/cost and indoor comfort; this in turn influences occupant behavior, thus forming a closed loop. Therefore, in the study of building energy consumption, it is difficult to avoid the study of human behavior in buildings. For example, Yan Zhang et al. clustered research keywords on human behavior and presented them with visual images, pointing out the close relationship among building energy consumption, human behavior, and environmental comfort [8]. Qindi Li et al. studied the influence of human behavior in the construction of a low-carbon campus in Shanghai, and the results showed that individual behavior directly affected the overall energy consumption and carbon emissions of the campus. Although the relevant energy-saving policies were introduced, energy consumption still increased by 5% per year [9]. Hoes P. et al. simulated human behavior in buildings and pointed out that architectural design should be optimized according to the actual human behavior and architectural characteristics for special buildings, and the description of human behavior should be more detailed [10]. Ioannou A. et al. analyzed the Monte Carlo sensitivity analysis results of factors affecting annual heating energy consumption and predicted mean vote (PMV) comfort index (related to human behavior) in Northern Europe [11]. Chuang Wang presented the environment- and event-related driving forces behind lighting usage in office buildings, described the degree to which these factors influence lighting use, and used a function and probability relations to describe the random process of lighting use [12]. The problem of energy consumption data quality cannot be ignored. In fact, in other related fields, some scholars have also conducted relevant research, for example, Gang Zhou et al. developed an iterative online fault identification framework utilizing a novel lost data repair technique [13].

To obtain the energy consumption monitoring data for guiding the research about human behavior and energy consumption, it is necessary to accurately obtain the lighting power consumption and socket power consumption, which have an obvious correlation with human behavior, as shown in Figure 1. It is worth noting that most public buildings in cold and severely cold regions of China use municipal hot water for central heating, which is charged by area rather than volume used actually and does not consume electricity from lighting sockets. It means that each room in winter does not consume electricity from lighting sockets to create a thermal environment, and the environment is created using the radiator with the hot water prepared centrally. This means that the research about separation methodology focuses not on energy consumption of heating but on electricity consumption.

In the past decade, the Chinese government supported the establishment of an energy monitoring platform to monitor and analyze energy consumption data online, especially electricity consumption. The energy monitoring platform has become an important technical carrier for building energy efficiency [14,15], but in most of the existing building energy monitoring platforms, these two items of power consumption data are often difficult to obtain. First, due to cost constraints and construction age, the construction process of the building electrical system is not designed according to the correct design of electrical items. The monitoring branches of lighting, sockets, and air-conditioning, shown in red in Figure 2, are mixed. Second, due to the changes in the functions and partitions of a building, the distribution circuit scheme also changes accordingly. In this case, even if the lighting is separated from sockets in the design scheme, the air-conditioning consumption is often mixed into the socket consumption in the renovation process.

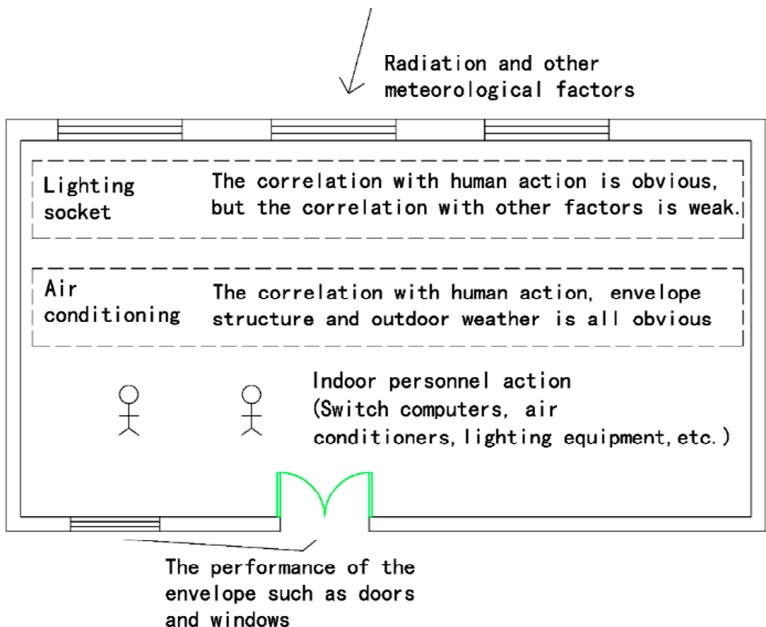

**Figure 1.** The indoor energy consumption component and its influencing factors in public buildings.

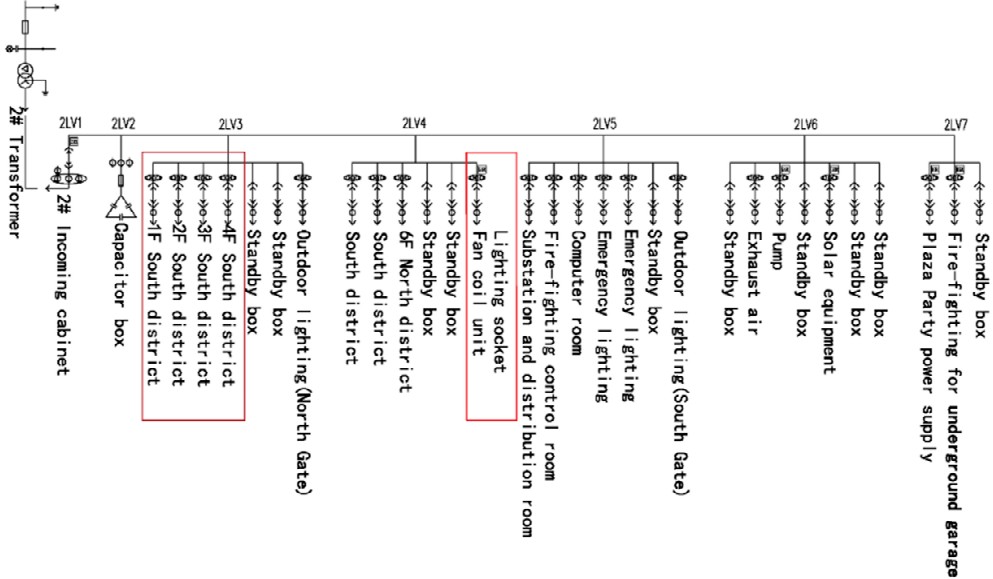

**Figure 2.** Construction drawing of electric energy monitoring for a sample building studied in this paper.

So, especially for public buildings, it is difficult to retrofit electric circuits massively, and at the same time, the aforementioned problems arise when the room function or room user changes. As a result, with this background, the itemized energy consumption data, which we obtain using an electrical loop monitoring instrument, will be inaccurate, because we fail to completely separate the power consumption of the lighting, socket, and air-conditioning terminal. Taking the study of human behavior as an example, if the power consumption of lighting sockets and air-conditioning cannot be accurately separated, it is difficult to evaluate the role of human behavior, outdoor weather, and envelope structure.

Therefore, how to separate the mixed power consumption of lighting sockets and air-conditioning without large-scale reconstruction is worthy of investigation. Scholars have presented a type of non-intrusive load monitoring (NILM) method. By measuring the curve of real-time power consumption, the start and stop electrical signals (current, voltage, active power, reactive power, power factor, etc.) of the equipment can be identified

for abnormal power consumption data identification and separation [16–18]. However, these relevant studies mainly focus on residential buildings with a few types of equipment. A possible reason for this is that the types of equipment in public buildings are complicated and their number is extremely large. Thus, it is extremely difficult to judge the state of equipment only by the total energy consumption curve. Therefore, Akbari [19] developed an end-use disaggregation algorithm based on the hybrid technology of simulation software and statistical analysis. This algorithm can adjust the estimation results according to the error size and can be used to separate the electrical energy consumption of multiple buildings in the same power grid, which overcomes some disadvantages of the NILM method. Newer studies, such as that of Li et al., used the outlier detection method to eliminate abnormal energy consumption data and then used canonical variable analysis to classify and predict energy consumption [20]. Wang proposed an energy consumption separation algorithm for an equipment load rate model. The estimation results of the equipment power consumption were obtained by inputting the rated power of the equipment, which means that the equipment power consumption is separated from total power consumption [21]. Doherty proposed a socket power consumption model, which was developed by comparing power consumption data and the corresponding equipment information. This model is applicable in separating various types of power consumption in sockets, but it is difficult to separate the air-conditioning power consumption, which is mixed with socket power consumption [22]. Anand proposed a new parameter, energy consumption per capita (*K*), to explain the stochastic relationship between energy consumption and utilization, which was developed by multiple nonlinear regression and deep neural network algorithms. This model can be used to estimate the energy consumption of different sub-items according to human behavior, which means that the sub-item power consumption is separated from total power consumption [23]. However, the common point of these energy separation methods [24–30] is to obtain the actual energy consumption data or to estimate the energy consumption of some sub-items according to the detailed information of buildings, equipment, and human behavior. The energy consumption data of different equipment can then be calculated, which means that this type of equipment power consumption is separated from total power consumption. If detailed information on buildings, equipment, and human behavior cannot be obtained, or is for public buildings, of which equipment and human behavior are complex, these methods still have some defects.

From the literature [1–7], we can discover the necessity of building energy efficiency, especially for public buildings. Therefore, how to use energy consumption data in monitoring platform is important, such as studying human behavior for an energy retrofit shown in the literature [8–13]. This means that accurate itemization measurement is important in the research of public building energy efficiency. However, in the face of various practical problems, it is often difficult to make accurate itemization measurement. Therefore, researchers have developed various algorithms to repair energy consumption data, such as those found in the literature [16–23], but most of the above separation methods rely on the detailed information of buildings, equipment, and human behavior. It is usually time-consuming and inefficient to survey this information. Thus, under the condition of generally abnormal energy consumption monitoring data and high labor cost, research on separation methodology, which is driven only by historical data and applied online with a universality of multiple scenarios without detailed information of buildings and equipment, is critical. Therefore, this study proposes an online separation methodology for separating the power consumption of lighting sockets and air-conditioning in public buildings. This methodology is driven only by historical data, without detailed information of buildings and equipment.

The rest of this paper is organized as follows. Section 2 introduces the basic information in this research, including the sample campus, building energy monitoring platform, and actual problems with data quality with two aspects. Section 3 introduces methods of pre-processing power consumption data with the k-means clustering algorithm. Section 4

introduces the methods and cases for identifying abnormal conditions for daily data and hourly data, compares the identification effect of the OATPM method and the KNN method, and then introduces the method and case for separating abnormal data after identifying data. Section 5 analyzes the applicability of the separation methodology. Section 6 discusses the selection of important parameters in the clustering algorithm. Finally, conclusions are presented in Section 7.

## 2. Background and Data for Power Monitoring

In this research, an office building in Dalian (referred to as Building A), an office building in Jinzhou (Building B), and a commercial building in Anshan (Building C) are used as the research sample set. Table 1 shows the basic information, Figures 3 and 4 the temperature information.

**Table 1.** Basic information for Buildings A, B, and C.

| Building Codes | Floor Height (m) | Floor Area (m²) | Energy Intensity (kWh/m²·Year) | Climate Partition | Heating Scheme | Cooling Scheme |
|---|---|---|---|---|---|---|
| Building A | 11.8 | 37,593 | 13.9 | Cold region | Municipal water | Separate air-conditioning |
| Building B | 56 | 26,000 | 46.6 | Severely cold region | Municipal water | Separate air-conditioning |
| Building C | 41 | 100,000 | 93.9 | Severely cold region | Municipal water | Fan coil unit |

Annotation: China can be divided into five climatic zones: severely cold region, cold region, hot-summer and cold-winter region, hot-summer and warm-winter region, and moderate region.

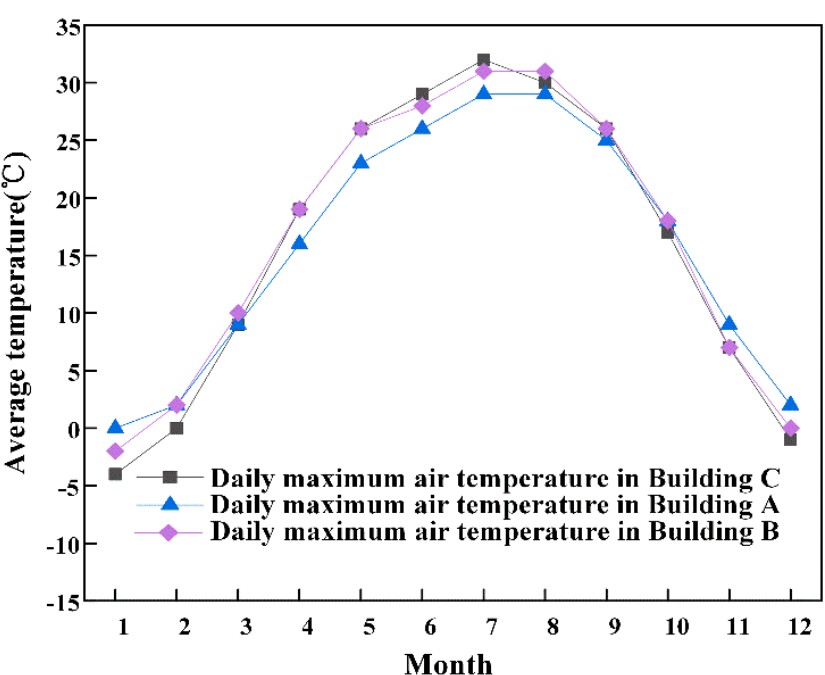

**Figure 3.** Daily maximum air temperature for three cities.

**Table 2.** Basic information on building energy consumption monitoring platforms.

| Energy Consumption Monitoring Platform | Building Sample | Total Number of Buildings (Blocks) | Floor Area of Buildings (×1000 m²) | Monitoring Point (Pieces) |
|---|---|---|---|---|
| Dalian public institution energy consumption monitoring | A | 15 | 300 | 951 |
| Liaoning public institution building energy consumption monitoring platform | B, C | 50 | 2030 | 1659 |

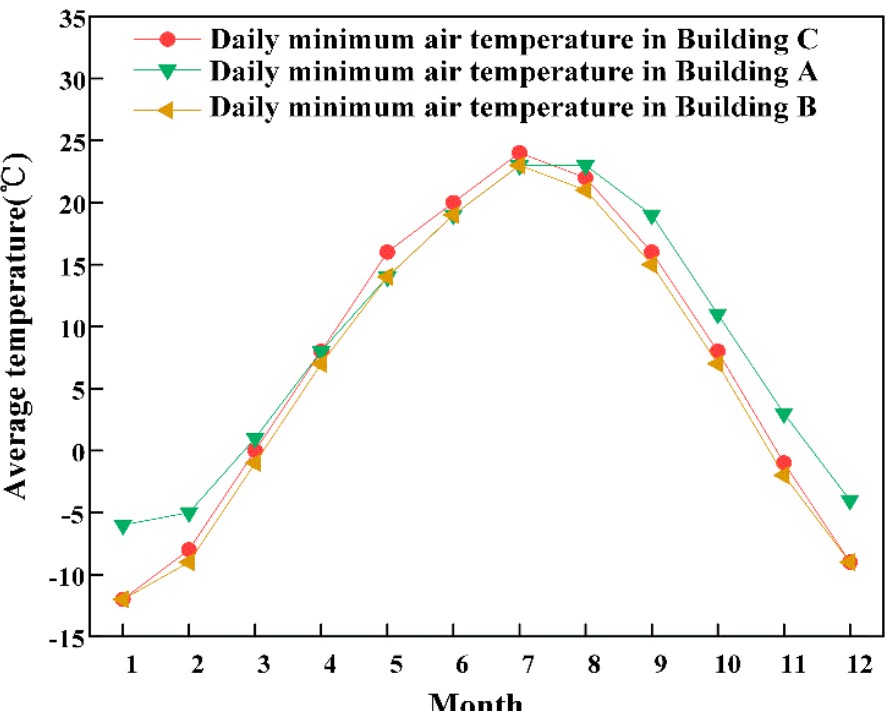

**Figure 4.** Daily minimum air temperature for three cities. The energy consumption monitoring platforms of Buildings A, B, and C are listed in Table 2.

While using the above platforms, there are many problems with data quality, which are also common problems of Chinese building energy consumption platforms, and which are reflected in the following two aspects.

(1) When problems occur in the process of data collection and transmission, they cannot be dealt with in a short period of time, resulting in missing data, data mutation, etc.

(2) When the energy measurement of low-power terminal equipment is ignored, or when different functions of low-power terminal equipment are collected by the same meter, the power consumption of lighting sockets and air-conditioning are mixed. For example, lighting, sockets, and air-conditioning are monitored by the same branch monitoring. In Building A, the peak value of lighting socket power consumption in the cooling season is 1.5 times that of the peak value in the transition season, as shown in Figure 5, which exceeds the reasonable power consumption range of the lighting socket equipment. This indicates that the data of lighting socket power consumption is mixed with the data of air-conditioning power consumption.

Therefore, cleaning abnormal data and separating mixed data is important for improving data quality, and it is also a crucial link for energy efficiency retrofits. There are two main dimensions to cleaning and separating abnormal data: one is for abnormal daily data, and the other is for abnormal hourly data. For daily data, the granularity of measurements is one day, and it means one day is one point. For hourly data, the granularity of measurements is one hour, and it means one hour is one point. Different methods are proposed to address the two dimensions, including the OATPM and KNN methods, and the details are in Section 4.

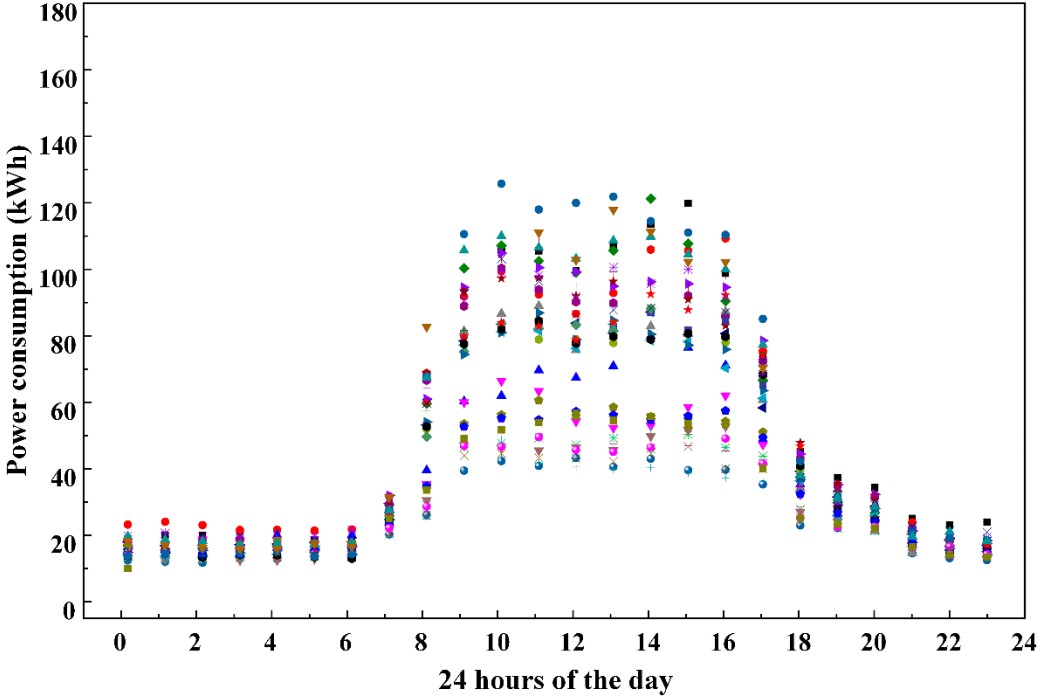

(**a**) Hourly data in transition season

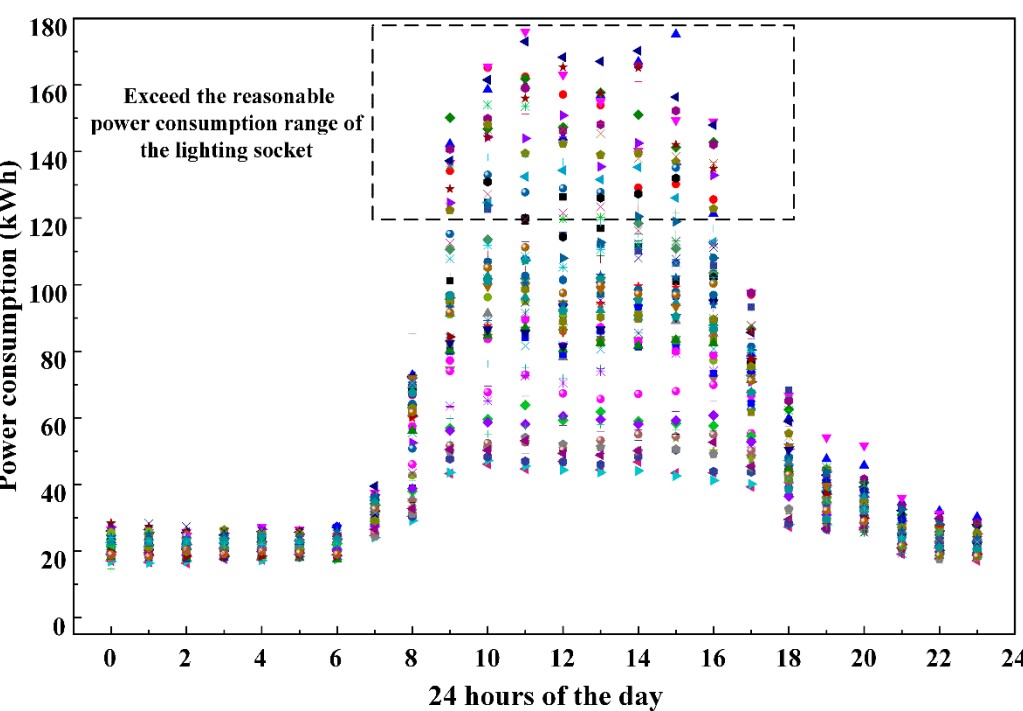

(**b**) Hourly data in cooling season

**Figure 5.** Time-series characteristics of the electricity consumption of lighting sockets.

## 3. Pre-Processing of Power Consumption Data

It is necessary to identify and clean abnormal data before separating the power consumption data of lighting sockets and air-conditioning. The abnormal conditions mainly include the data missing and data mutation storage. Among them, the values of missing data are displayed as "0," which means that they are easy to identify and clean. As

for data mutation, the partial power consumption data are too big or small, exceeding the actual threshold of energy consumption, as shown in Figure 5 and its description above. In this paper, the k-means clustering algorithm [30–34] is adopted to identify and clean the mutated data. The steps are as follows.

1.  The K sets of points are randomly selected as the initial clustering center in the sample data set. According to the three kinds of air-conditioning state (closed/slightly open/fully open) and three kinds of data state (located/above/below), K is evaluated at 3 for KNN throughout the paper.
2.  The distance between other points and the initial clustering center point is calculated, and these other points are allocated to the nearest neighbor cluster.
3.  After the preliminary clustering is completed, the averages of all sample points in different clusters are selected as the new clustering center, and then steps 1 and 2 are repeated.
4.  The clustering center and clusters of the sample points are updated iteratively, until the clustering center no longer changes, which means the end of this clustering algorithm. Next, we can output the clustering center and K pieces of the clusters of the sample points.

What needs illustration is that the distance between the points in step (2) can be calculated by the Minkowski distance [30–34], as shown in Equation (1), where $p$ is 2, when we compute the two-dimensional point, and $x_1$, $x_2$ are the value of the two points.

$$Dist(x_1, x_2) = \left[ \sum_{k=1}^{n} \left| x_{1,k} - x_{2,k} \right|^p \right]^{\frac{1}{p}} \tag{1}$$

Taking Building A as an example, workdays, weekends, and holidays are selected as different types, and abnormal data are identified and cleaned day by day, as shown in Table 3 and Figure 6. Holidays are different from weekends. Holidays mean rest days, which are usually for some festivals and for summer or winter vacations, such as Christmas vacation. If the number of samples in the largest and smallest cluster centers accounts for less than 5% of the total sample number, or if the difference between the cluster centers is too large, it will be judged as a data mutation.

**Table 3.** K-means clustering results of daily lighting socket consumption.

| State | Workdays | | | Holidays | | |
|---|---|---|---|---|---|---|
| The number of the cluster | 1 | 2 | 3 | 1 | 2 | 3 |
| Cluster center (kWh) | 1848.65 | 1263.52 | 485.15 | 1037.01 | 778.52 | 151.75 |
| Sample numbers (pieces) | 8 | 130 | 7 | 21 | 51 | 3 |
| Proportion (%) | 4.88 | 90.85 | 4.27 | 28.00 | 68.00 | 4.00 |
| State conclusion | Abnormal | Reasonable | Abnormal | Reasonable | Reasonable | Abnormal |

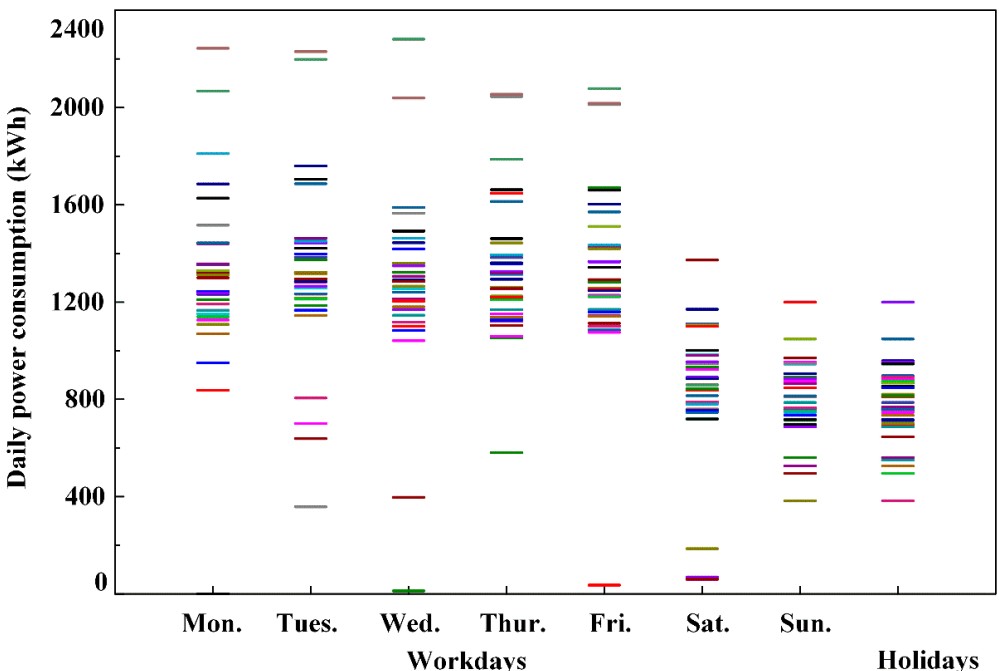

(**a**) Before cleaning abnormal data

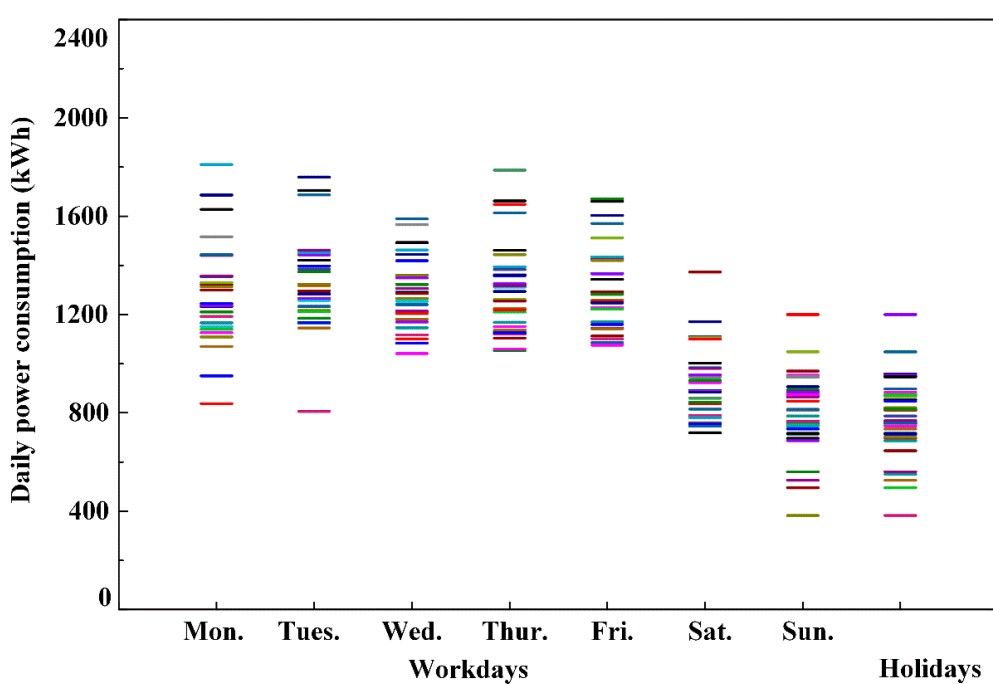

(**b**) After cleaning abnormal data

**Figure 6.** Scatter diagram of daily electricity consumption of lighting sockets in Building A.

From Figure 6, it is seen that the k-means method of cleaning daily data screens abnormal data effectively. The results of cleaning the abnormal hourly data in Building A are shown in Figure 7.

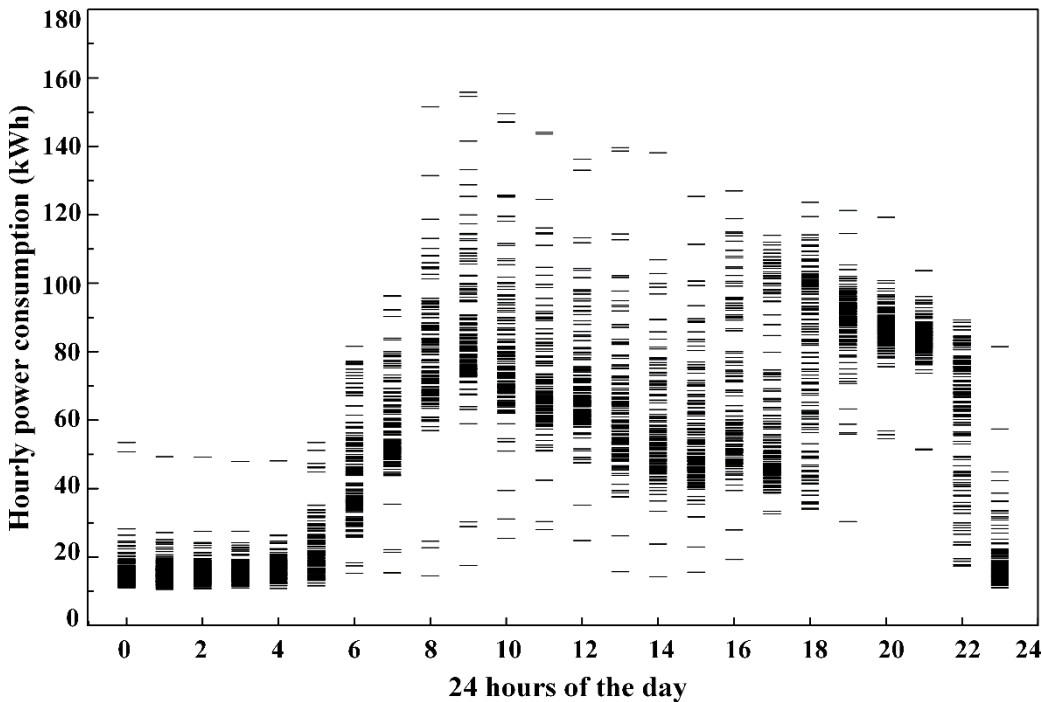

(**a**) Scatter diagram of daily power consumption before cleaning abnormal data

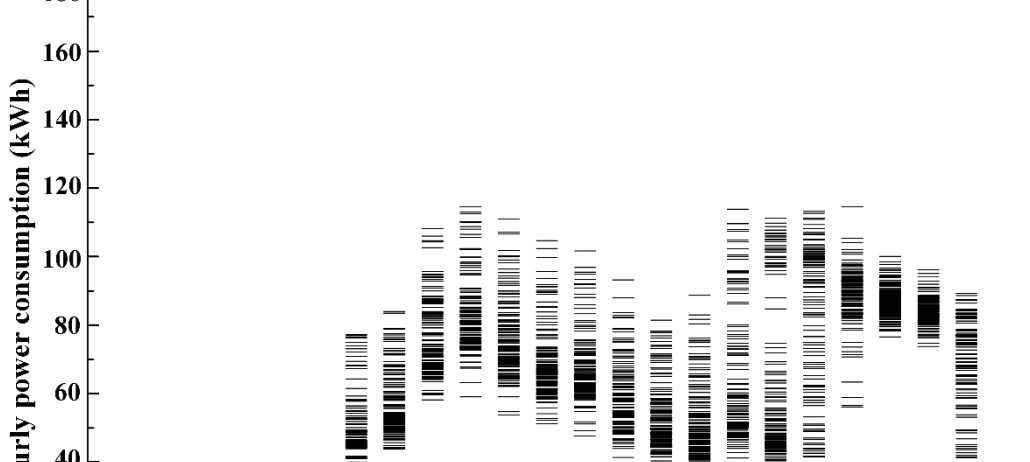

(**b**) Scatter diagram of daily power consumption after cleaning abnormal data

**Figure 7.** Scatter diagram of hourly electricity consumption of lighting sockets in Building A.

## 4. Separation Methodology

The proposed methodology needs to identify the abnormal days or hours first, and then use historical data to separate abnormal data. There are five main steps, as shown in Figure 8. The details are shown in the other sections that follow.

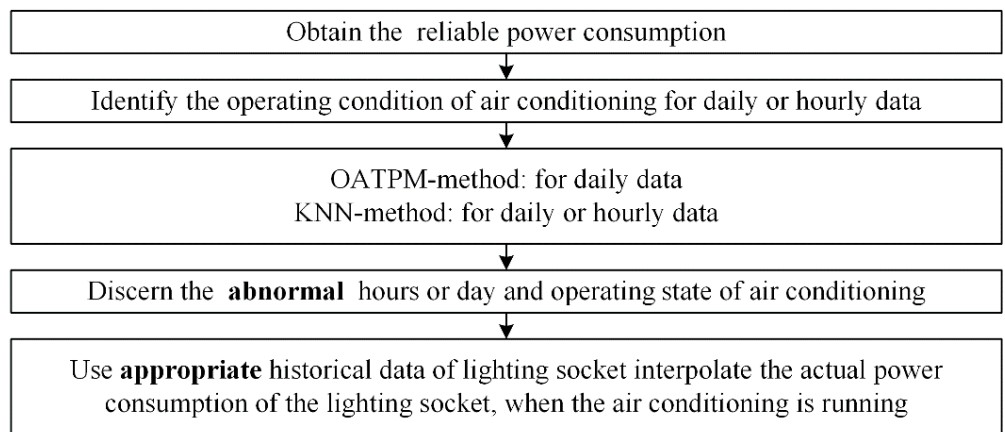

*Abnormal data, days or hours: data, days or hours with mixing the power consumption of the air conditioning*
*Appropriate data: without mixing the power consumption of the air conditioning*

**Figure 8.** Diagram of the overall model of implementing the identification and separation method.

For the ways to discern the abnormal hours or day and operating state of air-conditioning, mentioned in Figure 8, there are three main steps as follows:

- Use the model of average outdoor temperature and lighting socket daily power consumption, or the k-means clustering algorithm, to obtain the eigenvalues of the method, including clustering center (c), outdoor temperature threshold ($T_{Min}$), and power consumption data threshold ($\overline{E}_{Min}$), etc.
- Use the above eigenvalues and the lighting socket power consumption with air-conditioning opened to identify the state of the air-conditioning.
- Use appropriate historical data of lighting sockets (without mixing the power consumption of the air-conditioning) to predict and interpolate the actual power consumption of the lighting socket when the air-conditioning is running.

The OATPM method is appropriate for identifying the state of air-conditioning for daily data, because it can only identify the closed or open state. However, the KNN method is appropriate for identifying the state of air-conditioning for daily data or hourly data, because it can identify the closed, slightly open, or fully open states.

### 4.1. Method for Identifying Abnormal Conditions for Daily Data

According to the ratio of air-conditioning running hours ($p$), the state of air-conditioning is specified. If $p < 40\%$, it is judged that the air-conditioning was closed; if $40\% < p < 70\%$, it is judged that the air-conditioning was slightly open; if $p > 70\%$, it is judged that the air-conditioning was fully open. The steps of these two methods are shown in Figure 9a,b.

It is worth noting that the KNN method also uses the model of average outdoor temperature and daily lighting socket power consumption for the eigenvalues (cluster center and maximum and minimum values of the cluster interval, such as $c$, $T_{min}$, $\overline{E}_{min}$) whether identifying abnormal daily or hourly data. Thus, it will not repeat the same process in Sections 4.1–4.5.

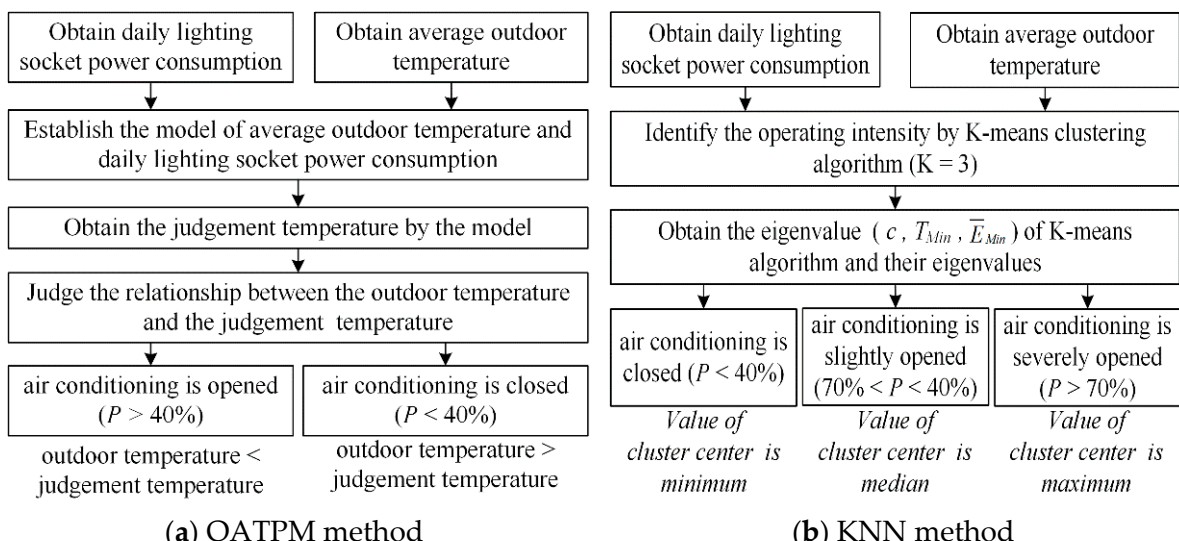

**Figure 9.** Identification method of air-conditioning usage condition on the designated day.

### 4.2. The Case of Identifying Abnormal Conditions for Daily Data

From April to August 2018, Building A is selected to verify the results for identifying and separating the normal data by the above methodology. The parameter *a* is used to evaluate the results, as follows:

$$a = \frac{n}{N} \tag{2}$$

For the OATPM method, taking daily power consumption as the ordinate and the daily average outdoor temperature as the abscissa, the model of daily average outdoor temperature and daily power consumption of lighting sockets (mixed with power consumption of air-conditioning) is shown in Figure 10. The three clusters are displayed in three colors.

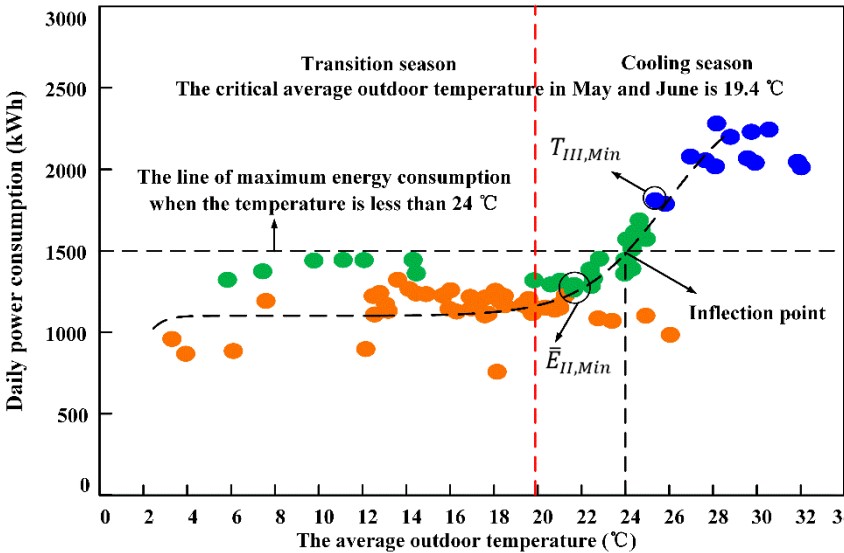

**Figure 10.** Model of average outdoor temperature versus daily electricity consumption.

The data are fitted into a curve, and the second derivative is calculated to obtain the inflection point (24 °C, 1445 kWh). The average outdoor temperature and power consumption are, respectively, represented by black dotted lines. As shown in Figure 10, when the outdoor temperature is lower than 24 °C the daily power consumption is within the range of 750–1500 kWh, and the maximum value is 1445 kWh.

When the outdoor temperature is higher than 24 °C, the daily cumulative electricity consumption is higher than 1445 kWh. According to the properties of the second derivative, at 24 °C, the plus and minus signs of the first derivative change, that is, the local extreme value appears. Therefore, 24 °C is presumed to be the point of the power consumption characteristic change. At the same time, analyzing this point by the correlation coefficient, the power consumption of the lighting socket equipment is weakly correlated with the outdoor temperature, but the power consumption of the air-conditioning is strongly correlated with outdoor meteorological conditions. The conditions when $\overline{T_i}$ is higher than 24 °C and $\overline{T_i}$ is lower than 24 °C are calculated, respectively, about the correlation coefficient between the average outdoor temperature and the daily power consumption, as follows:

$$r_{XY} = \frac{cov(X_i, Y)}{\sqrt{var(X_i)var(Y)}} \tag{3}$$

where $X_i$ is the average daily outdoor temperature and $Y$ is the power consumption of the lighting socket (mixed with the power consumption of air-conditioning). The value of the correlation coefficient $R$ ranges from 0 to 1. The closer it is to 1, the stronger the correlation between variables; the closer it is to 0, the weaker the correlation. After the calculation, $R(\overline{T_i} \geq 24\,°\text{C}) = 0.709$; $R(\overline{T_i} \leq 24\,°\text{C}) = 0.196$.

Therefore, when the average outdoor temperature is lower than the judgment temperature (24 °C), the power consumption is small, and the relationship between the average outdoor temperature and the lighting socket power consumption is relatively weak, so it can be considered that the air-conditioning power consumption is not mixed with the lighting socket power consumption. When the average outdoor temperature is higher than the judgment temperature (24 °C), the power consumption increases suddenly, and the maximum value of the power consumption curve appears. The relationship between the average daily outdoor temperature and the power consumption is relatively large. Thus, it can be considered that the air-conditioning power consumption is mixed with the lighting socket power consumption.

Regarding the KNN method, it is calculated by the k-means algorithm, of which the parameter $K$, the number of clusters, is also evaluated at 3, as discussed above. That is, closed ($p < 40\%$), slightly open ($40\% < p < 70\%$), and fully open ($p > 70\%$). The three types of cluster distributions are also marked in three different colors, as shown in Figure 10.

The identification results with different methods are calculated for analyzing the advantages and disadvantages of the two proposed methods, as shown in Figure 11a,b. The accuracy a of identification results for the OATPM method is 83.9%, and the accuracy for the KNN method is 93.1%. This shows that the accuracy of the OATPM method is slightly lower than that of the KNN method, but it still reaches a high level. At the same time, the OATPM method only needs to calculate the second derivative without a complex algorithm, so the calculation speed is high. A more detailed comparison is made in Table 4.

**Table 4.** Comparison of the OATPM and KNN methods.

| Method | Identification Accuracy | Applicable Occasions | Degree of Identification | Can It Be Used to Predict Energy Consumption? |
|---|---|---|---|---|
| Temperature model | 83.9% | For daily data | Closed/open | Yes |
| Clustering | 93.1% | For daily data | Closed/slightly open/fully open | No |
| algorithm | 96.2% | For hourly data | | |

Note: The temperature models can be used to predict energy consumption.

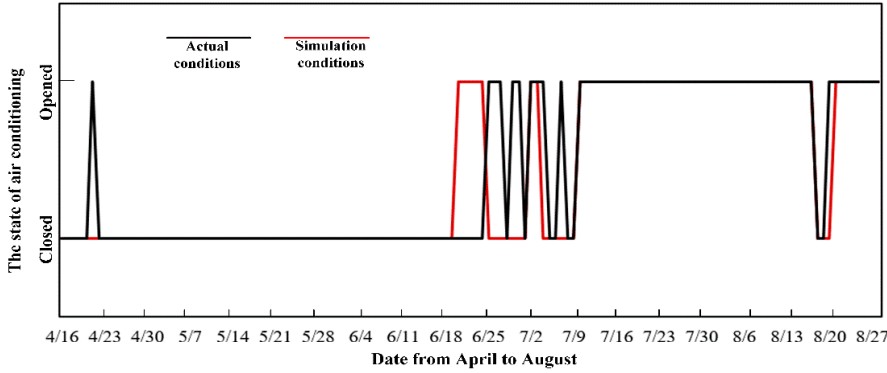

(**a**) Comparison diagram between the simulation and actual conditions using the OATPM method.

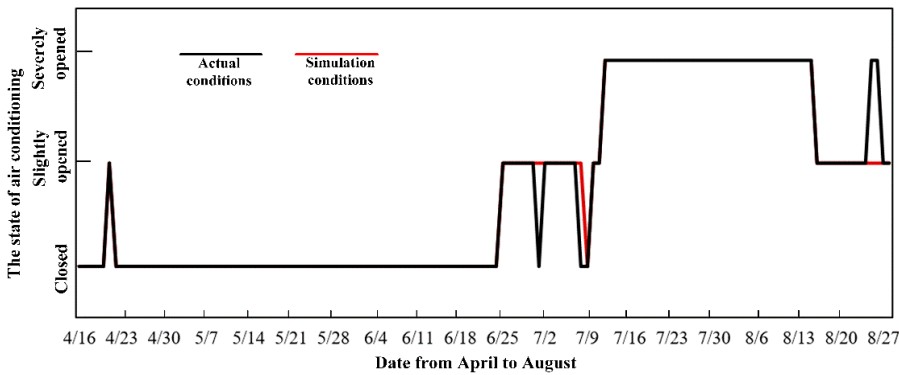

(**b**) Comparison diagram between the simulation and actual conditions using the KNN method.

**Figure 11.** Comparison of identification results using different methods.

### 4.3. Method for Identifying Abnormal Conditions for Hourly Data

According to the above analysis, only the clustering algorithm is suitable for this situation, of which the steps are similar to those shown in Figure 9b. The steps of this method for hourly data are not exactly the same as the steps for daily data.

(1) Using the clustering algorithm to generate the critical eigenvalues, $T_{III, Min}$ and $\overline{E}_{II,Min}$, when the power consumption and the outdoor temperature are less than $T_{III,Min}$ and $\overline{E}_{II,Min}$, respectively, the state of the air-conditioning is closed. For the contrary state, the air-conditioning is open.

However, the state of the air-conditioning does not completely depend on the outdoor climate, and human behavior is also a type of complex influencing factor. For example, when the power consumption is higher than $\overline{E}_{II,Min}$ and the outdoor temperature is less than $T_{III,Min}$, it is possible that air-conditioning is closed and the lighting socket power consumption is higher, as shown in Figure 12 (the fuzzy area in the upper left corner). When the outdoor temperature is higher than $T_{III,Min}$ and the power consumption is less than $\overline{E}_{II,Min}$, it is possible that people do not use air-conditioning due to their habits, even if the outdoor temperature reaches the degree of using air-conditioning, as shown in Figure 12 (the fuzzy area in the bottom right corner). Therefore, after the end of step (1), the next step should be identified.

(2) The theoretical maximum power consumption $e_{i,Max}$ of the lighting socket in this special hour is calculated and compared with the actual power consumption $e_i$. If $e_i$ is higher than $e_{i,Max}$, it is considered that the power consumption of the lighting socket is mixed with that of the air conditioner; otherwise, it is not mixed.

### 4.4. The Cases of Identifying Abnormal Conditions for Hourly Data

The data of Building A in August 2018 are selected for the case analysis, as shown in Figure 12. The three cluster centers and eigenvalues, such as large and small thresholds, are shown in Table 5.

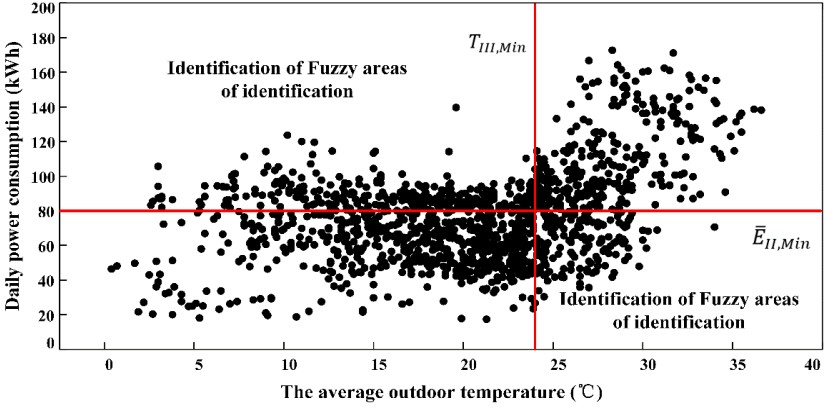

**Figure 12.** Characteristic value recognition results.

**Table 5.** Eigenvalues of three clusters by the KNN method.

| Cluster Categories | Number of Samples | Eigenvalues | Power Consumption of Lighting (kWh) | Outdoor Temperature (°C) |
|---|---|---|---|---|
| Cluster I: closed | 47 | C | 1152.50 | 17.28 |
| | | MAX | 1282.65 | 24.55 |
| | | MIN | 805.47 | 3.30 |
| Cluster II: slightly open | 27 | C | 1448.11 | 19.21 |
| | | MAX | 1686.75 | 24.83 |
| | | MIN | 1287.33 | 19.89 |
| Cluster III: fully open | 13 | C | 2066.55 | 28.87 |
| | | MAX | 2280.90 | 32.05 |
| | | MIN | 1787.04 | 25.38 |

The comparison diagram between the simulation conditions and the actual conditions for hourly data is shown in Figure 13, with 93.1% accuracy. It is divided into three states (closed, slightly open, and fully open) for daily data, but this is not necessary for the complex steps for hourly data. It can also be divided into two states (closed and open) without affecting the results.

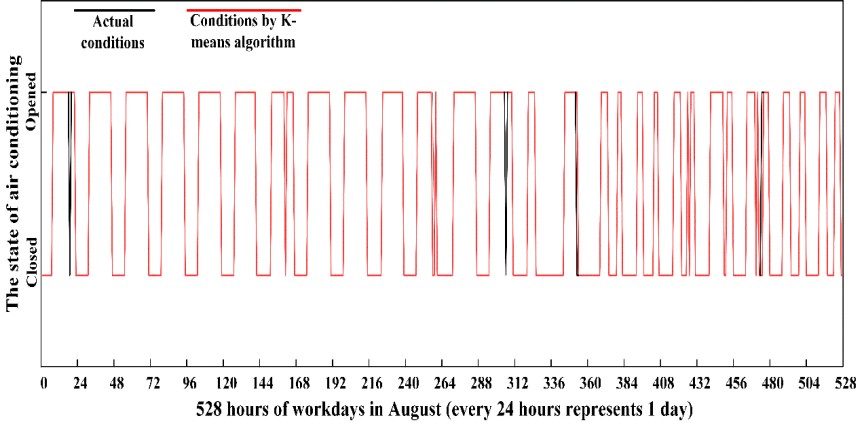

**Figure 13.** Comparison of identification results using the KNN method.

### 4.5. Comparison of the OATPM Method and the KNN Method

Together, the OATPM method and the KNN method constitute the online identifying methodology, which is the first step of the separation methodology. A comparison of their advantages and disadvantages is displayed in Table 5. According to different application conditions, different methods can be selected to identify the air-conditioning state, to better separate the mixed data.

### 4.6. The Method and Case of Separating Abnormal Data after Identifying Data

After identifying the normal data and their corresponding hours or days, the work of separating abnormal data begins.

Abnormal data appear on abnormal days or hours, which need to be separated. The normal power consumption for the nearest days or hours of the abnormal days or hours are selected for interpolation. If the nearest data are also abnormal, the second-nearest data need to be selected for interpolation. By analogy, the normal power consumption data of lighting sockets without air-conditioning can finally be obtained.

Equation (4) is used to calculate the relative error between the lighting socket power consumption of the separation method and the actual situation, as follows:

$$\delta = \frac{\Delta}{p_{i,j}} \times 100\% = \frac{|e_{i,j} - p_{i,j}|}{p_{i,j}} \times 100\% \tag{4}$$

where $\delta$ is the relative error when separating mixed data (%), $e_{i,j}$ is the lighting socket power consumption of the separation method (kWh), and $p_{i,j}$ is the lighting socket power consumption of the actual situation (kWh). The data of Building A from April to August 2018 are used for verifying the results of the separation. The error of 355 data points is shown in Figure 14, and some parts of the details are shown in Figure 15.

It is seen from Figure 15 that the relative error when separating mixed data is mostly within 10%. The numbers of samples for which the relative error is more than 10% accounts for 3.94%. Therefore, this method is feasible to a certain extent.

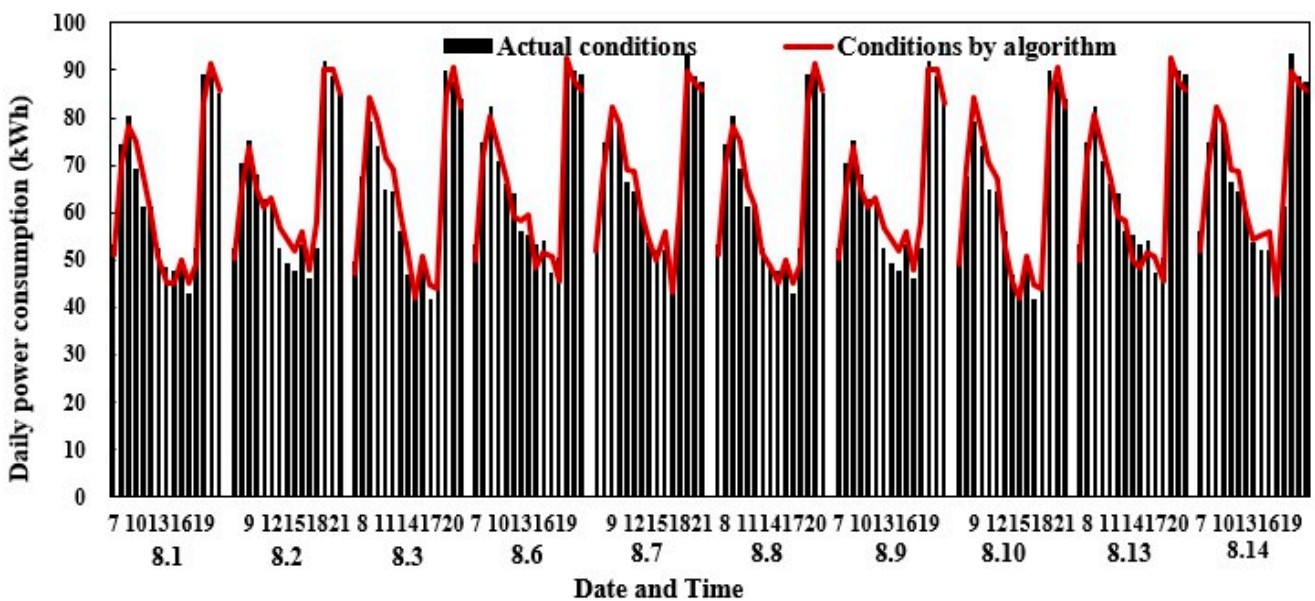

**Figure 14.** Comparison of separating the mixed data of the separation method and actual data.

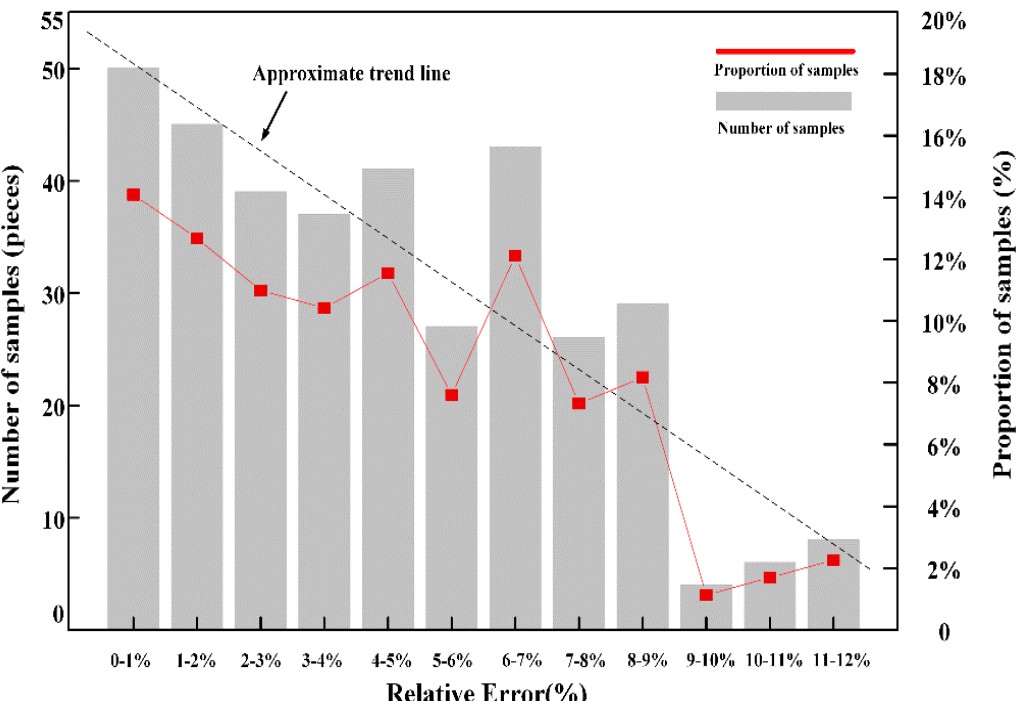

**Figure 15.** Relative error number distribution when separating mixed data.

### 5. Discussion of the Separation Methodology

The human behavior, type of equipment, climates, and functions of different buildings are different, which leads to obvious differences in the power consumption characteristics of different buildings. To verify the applicability of the methodology for identifying and separating mixed data, on the premise of the successful verification of Building A, Buildings B and C, having different areas and functions, are selected as samples for further verification.

The results show that the methodology is also applicable to buildings with other functions and climates, of which the identification accuracy is above 90%. A comparison of identifying and separating data is shown in Figure 16 and Table 6. As seen in Figure 16, the trend of separation is basically consistent with the actual situation, and the separation power consumption is close to the actual value.

The relative errors of separation in Buildings A, B, and C are shown in Figure 17. The relative error of buildings having different functions is within 15%, and the relative error less than 10% in each building is more than 85%, which indicates the high applicability of the proposed method.

**Table 6.** Accuracy of identifying operating conditions of air-conditioning.

| Building | Climate Zone | Building Function | Working Time | Form of Air-Conditioning | Accuracy | |
|---|---|---|---|---|---|---|
| | | | | | For Daily Data (%) | For Hourly Data (%) |
| Building A | Cold | Office building | 7:00–22:00 | Separate air-conditioning | 93.1 | 96.3 |
| Building B | Severely cold | Office building | 7:00–17:00 | Separate air-conditioning | 91.2 | 93.4 |
| Building C | Severely cold | Commercial building | 8:00–21:00 | Fan coil unit | 95.6 | 96.0 |

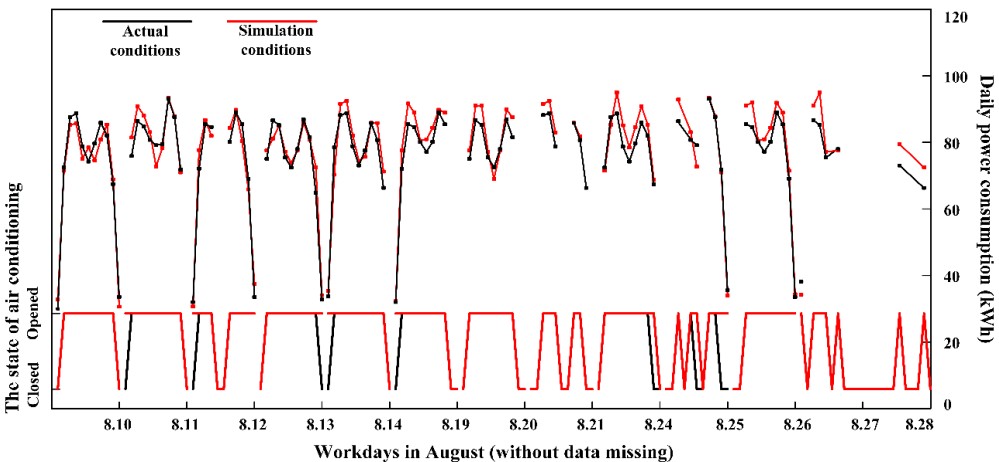

(**a**) Comparison of identifying and separating abnormal data in Building B

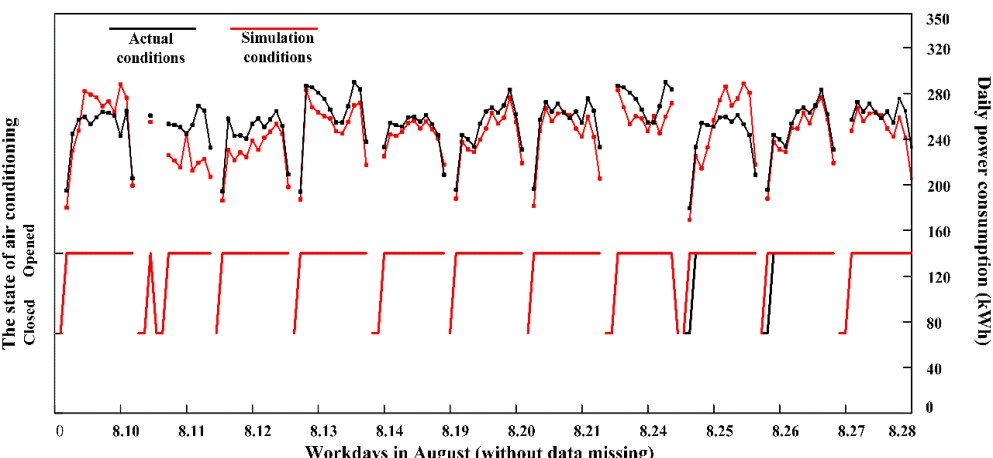

(**b**) Comparison of identifying and separating abnormal data in Building C

**Figure 16.** Comparison of identifying and separating abnormal data by the proposed method.

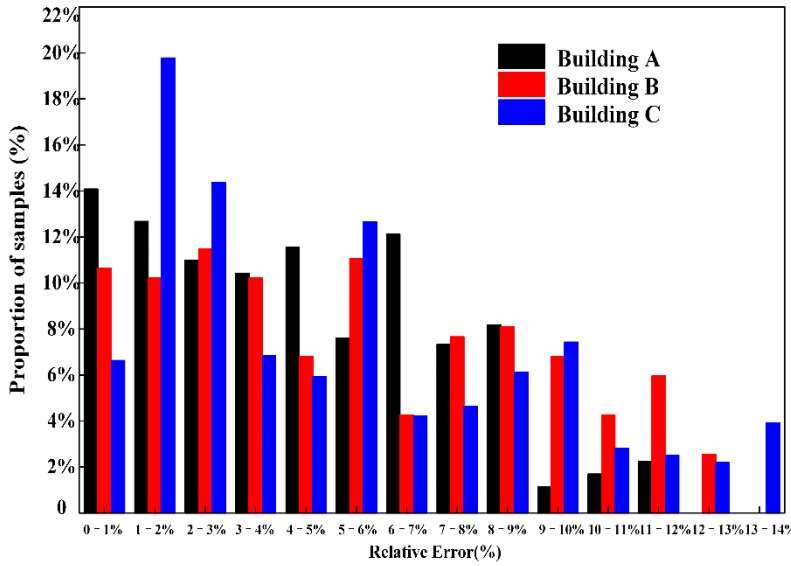

**Figure 17.** Distribution of the relative error between the separation data and the actual values.

At present, there is no accurate specification for energy consumption data processing, but there are clear specifications in the related field of energy consumption prediction, as shown in Table 7 [35,36]. From the strictest point of view, the error of this method is within the acceptable range.

**Table 7.** Relevant international regulations.

|  | Technical Code in China (JGJ176-2009) | IPMVP | FEMP |
|---|---|---|---|
| Error in one month | ±15% | ±20% | ±15% |
| Error in one year | No Standard | No Standard | ±10% |

Note: Technical code in China (JGJ176-2009) is "Technical code for the retrofitting of public building on energy efficiency"; IPMVP is "International Performance Measurement and Verification Protocol"; and FEMP is "US Federal Energy Management Program".

For future research, on the one hand, the results of the study show that the effect is obvious in cold regions and severely cold regions. As this method is purely data-driven, its applicability to other climatic zones is of value for further study. On the other hand, the kind of methodology that is driven by historical data with nearest-neighbor clustering algorithm is also widely cited in other fields about building energy efficiency—for example, the research about the effects of human behavior or meteorology on energy consumption. So, it also has reference value for the research in other fields.

## 6. Selection of Important Parameters in the Clustering Algorithm

In the process of identification and separation by the KNN method, the selection of some parameters is critical for accuracy, primarily the selection of clustering samples and the number of clusters, *K*.

As for the number of clusters, *K*, it should be selected according to actual demand. When one needs to identify and separate abnormal data by the KNN method for daily data, the three states, that is, closed, slightly open, or fully open, should be considered. When one needs to identify and separate abnormal data for daily data, it is enough to divide into two states, that is, closed and open.

Regarding the selection of clustering samples, the number of clustering samples, such as samples in a whole year or in a cooling season, directly affects the clustering results. In severely cold and cold climate areas, outdoor temperature values vary greatly in different seasons. Thus, in the case of clustering with outdoor temperature and power consumption values as variables, the outdoor temperature values vary with the number of clustering samples, which means that there is a difference between the identification and separation results. To obtain the best size of samples with strong applicability and more accuracy, the sample in a whole year, a cooling season, or a "transition season + cooling season" are selected for validation.

Samples for a whole year of Building A are selected for identifying and separating using the k-means clustering algorithm. The detailed information for these samples is shown in Table 8.

According to the data in Table 7, the following problems are noted. First, the variation trend of power consumption and the outdoor temperature are different, which means that identifying the three states (closed, slightly open, fully open) is difficult. Second, it is difficult to guarantee the eligible identification accuracy. If the power consumption value is used to define the three states of air-conditioning, the accuracy of identification results is 73.25%. If the outdoor temperature value is used to define the three states of air-conditioning, the accuracy of identification results is 22.42%. Finally, the outdoor temperature threshold difference of each cluster is too large, which means that identifying the state of air-conditioning is difficult.

**Table 8.** Detailed information of the samples for a whole year by KNN method.

| Cluster Categories | Number of Samples | Eigenvalues | Power Consumption of Lighting (kWh) | Outdoor Temperature (°C) |
|---|---|---|---|---|
| Cluster I: closed | 124 | C | 1182.52 | 16.00 |
| | | MAX | 1329.90 | 24.95 |
| | | MIN | 805.47 | −8.72 |
| Cluster II: slightly open | 74 | C | 1517.24 | 2.52 |
| | | MAX | 1787.04 | 30.47 |
| | | MIN | 1353.89 | −12.10 |
| Cluster III: fully open | 25 | C | 2089.84 | 29.09 |
| | | MAX | 2280.90 | 32.05 |
| | | MIN | 1610.49 | 25.38 |

Similarly, samples of the cooling season (from June to August) for Building A, and samples of the "transition season + cooling season" (from April to August) for Building A are also selected. The different results of identification are shown in Figure 18, with the detailed data shown in Tables 3 and 9 separately. It is seen that the identification results of air-conditioning with the data of "cooling season + transition season" is the closest to the actual conditions, followed by the results of the cooling season.

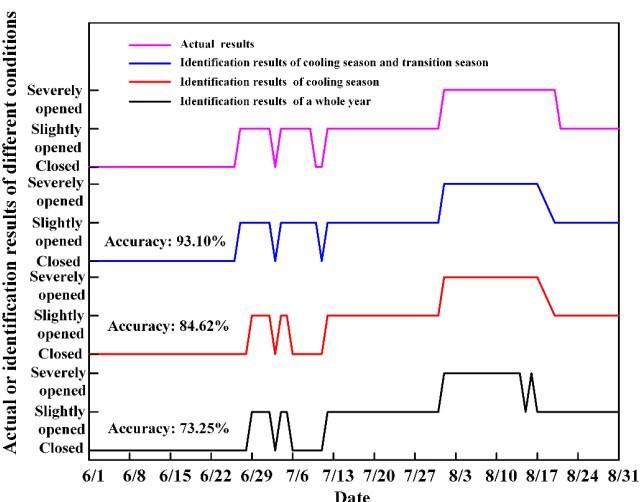

**Figure 18.** Recognition results of air-conditioning usage conditions with different variable durations.

**Table 9.** The detailed information of the samples for a whole year by KNN method.

| Cluster Categories | Number of Samples | Eigenvalues | Power Consumption of Lighting (kWh) | Outdoor Temperature (°C) |
|---|---|---|---|---|
| Cluster I: closed | 25 | C | 1190.13 | 20.79 |
| | | MAX | 1329.90 | 24.95 |
| | | MIN | 805.47 | 17.35 |
| Cluster II: slightly open | 17 | C | 1495.04 | 24.62 |
| | | MAX | 1686.75 | 30.47 |
| | | MIN | 1353.89 | 22.41 |
| Cluster III: severely open | 13 | C | 2066.55 | 28.84 |
| | | MAX | 2280.90 | 32.05 |
| | | MIN | 1787.04 | 25.38 |

## 7. Conclusions

(1) The OATPM and KNN methods are driven by the historical data of the energy monitoring platform, which can effectively separate the power consumption of lighting sockets and air-conditioning in public buildings. According to the three kinds of air-conditioning state (closed/slightly open/fully open) and three kinds of data state (located/above/below), $K$ is evaluated at 3 for KNN throughout the paper. The identification error for three public buildings utilizing the method was less than 15%, and the proportion of error greater than 10% was less than 15%.

(2) The OATPM method is suitable for identifying and separating daily data, and the calculation speed is high, but it cannot identify and separate hourly data. Thus, it is suitable for scenes with low identification accuracy, such as research on total energy consumption statistics. The KNN method is suitable for identifying and separating not only daily data but also hourly data; however, the calculation is complex and slow. Therefore, it is suitable for scenes with high identification accuracy, such as research on the correlation between human behavior characteristics and energy consumption.

(3) The methodology proposed in this study is suitable for public buildings with different functions and climates, especially for buildings with high power consumption values for lighting sockets or large differences between power consumption characteristics of air-conditioning and lighting sockets. For example, in commercial buildings, the power consumption of air-conditioning is relatively large. Therefore, the power consumption of lighting sockets is far less than the power consumption of lighting sockets (mixed air-conditioning) in the cooling season, which means that the number of clustering iterations is fewer, and the distances between different clusters is greater. Thus, the identification results of this separation methodology for commercial buildings are accurate.

**Author Contributions:** Methodology, validation, writing—original draft, T.Z.; investigation and modeling, C.Z. and T.U.; conceptualization and supervision, L.M. All authors have read and agreed to the published version of the manuscript.

**Funding:** This work was supported by the National Key Research and Development Project of China (Grant No. 2017YFC0704203), National Natural Science Foundation of China (Grant No. 52078096) and "the Fundamental Research Funds for the Central Universities" (Grant No. DUT20JC47).

**Institutional Review Board Statement:** Studies do not involve humans or animals.

**Informed Consent Statement:** Studies do not involve humans.

**Conflicts of Interest:** The authors declare no conflict of interest.

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
