# Peer review of "Online Methodology for Separating the Power Consumption of Lighting Sockets and Air-Conditioning in Public Buildings Based on an Outdoor Temperature Partition Model and Historical Energy Consumption Data"

_applsci, doi:10.3390/app11031031_

Round 1
Reviewer 1 Report
Focus: The authors proposed methodology for identifying the state and separating the mixed power consumption data of HVAC system and lighting sockets in public buildings.
Strong Points:
Well written and well organized
The authors tackled a very important research issue
Potential Improvements:
1- I did not see some comparison results. How do you compare with related work. For example, i found one related work "Buildings Energy Efficiency Analysis and Classification Using Various Machine Learning Technique Classifier"
https://doi.org/10.3390/en13133497
2. The authors did not mention paper break down which is usually mentioned at the end of Introduction section.
3. Related work section is missed.
Author Response
请参阅附件。

Reviewer 2 Report
Comment:
The manuscript shows promising idea considering introduced separation methodology for separating the lighting power consumption from the overall power consumption. The weakness of the research is that it shows development and application of the method on two parts of climate associated with cities where their method is analyzed (transition season and cooling season), and one considerable part is neglected – heating season. It is essential to provide additional analysis for heating season and either augment developed model or compare two equally important periods – “transition + cooling season” and “transition + heating season”. It is essential in order to prove the 4th statement in the conclusion (line 378 - 386). The heating season for cities from the manuscript is equally important as the cooling season if we observe building energy demands for a standard year (possibly it has even greater impact on energy consumption then cooling season).
If you register considerably different results when you include heating season in you research, you can rearrange the manuscript by limiting your findings just to cooling season.
Remarks:
Introduction
1. Line 35: Please describe an abbreviation PMV. It is not introduced before line 35 and it is not used later in the manuscript.
2. Line 39-41: Please describe the correlations between human behavior and lighting energy demand in more detail. In line 23 you stated that air-conditioning has the largest impact on a whole building energy demand – 43%, so you should say something about the impact of human behavior on it. Those percentages you mentioned in line 23 are for a littoral climate because analysis in the paper you referenced was done for Hong Kong. Climate in Hong Kong is not the same as climate for three cities you analyzed because you can register temperatures below 0 °C and minimum temperature in Hong Kong is in January and it is cca. 14 °C. For your cities, you will have energy demand for heating (i.e. power consumption in the heating season) in the same amount as energy demand for cooling (air-conditioning) so you must take this in concern when developing a model.
3. Lighting energy demand depends on a building type. E.g. it has large impact on a whole building energy demand if you observe a commercial building (as you stated), but it does not necessary has large impact on building energy demand for a residential building. For that reason, it is useful to define on which building your method will work.
4. Line 52: In Figure 1. in caption you have term “normal buildings”. It is not clear what are “normal buildings”. It matters if a building is residential, commercial, school, etc., if you observe energy performance of a building.
Research design
5. In your research you just observed a cooling season and a transition season. What about heating season? As I found, all three cities you analyzed have a heating season, and it is equally important as the cooling season. For the heating season, you will probably have similar results (if you compare it to a diagram on Figure 6. b) as for the cooling season if we observe power consumption.
6. Also, you should provide full year climate data for those three cities you observed (e.g. temperatures, solar radiation, etc. in a form of a figure) so a reader can get a point on which climate your method can be applied or is it climate independent (but for that you should include results from other cities with different climate because those three buildings you observed are in cities with similar climate conditions).
Methods
7. If you include the heating season in your research, model from a Fig. 11. will not have the same shape. I assume it will be U-shaped, probably mirrored over temperature 10 °C. This is essential as I stated before because you can register temperatures below 0 °C in analyzed cities and power consumption will certainly raise so the model from the Fig. 11. will be different.
8. Line 113 – Line 116: Text is centered. Please justify this part, so it follows journal template.
Overall, methods that were used in the manuscript are well defined and described.
Results
9. Results are well defined for described inputs. On the other hand, they must be extended according to changes in the model after the heating season is included in the research.
Conclusions
10. Line 365: You stated that your methodology is suitable for public buildings with different functions and climates. For that statement you should provide results for various climates, because in the manuscript you provided three cities with similar climate so this conclusion can not be confirmed with the results.
11. You can not state that “heating season + transition period” will have the same impact on power energy consumption as “cooling season + transition period” because you did not analyze it in the paper. You should include heating season in the results and compare those two cases. Heating season for your three cities is generally speaking from mid-October to mid-April.
Reviewer 3 Report
This paper is about estimating energy consumption in buildings without detailed information about buildings, equipment, and human behavior. This requires separating the sources of energy consumption, for which the paper proposes a computational methodology. The evaluation is on the data from three buildings in China.
Several strong points argue for considering this paper's acceptance. The paper addresses an interesting and practical problem, which is sufficiently motivated. The separation methodology appears to work, and the low-level technical choices seem sound. The description of the related work appears sufficient and reasonable. The accuracy of the results is promising.
The main downside of this paper is its confusing, difficult-to-read presentation and writing.
Data needs to be better described: what is the granularity of measurements? Is it one day = one point or one hour = one point? If the latter, then there is aggregation of data by days and it needs to be clearly stated, as well as the level at which cleaning for anomalies is done. What are the relevant dimensions of each data point? Later on we learn about weekdays and weekends, but it needs to be described in section 2. Is weekend = holiday?
Section 4 starts off in a way that makes it hard to follow: the textual description has 3 steps, but the figure 9 has 5 boxes. The two descriptions do not share common words and seem to describe two different processes. The similarities and distinctions between OATPM and KNN methods are not introduced appropriately, making the reader reconstruct them in their head. The subsections of 4 are not described up-front, and the reader does not know what to expect. It is never clearly said that the OATPM is for daily data and KNN is for hourly (is it true? I'm not sure). As a result of this, the technical contribution of this paper is difficult to follow and understand in detail, even though at a high level it makes sense.
The writing can also use some improvement. Some sentence structures are awkward and take some effort to follow. Many garden pass sentences and long noun phrases. Try putting no more than 2 words between the subject of the sentence and its verb.
Minor comments:
- the first few sentences of the introduction are not clear if they provide the numbers for China's or worldwide total consumption.
- afore-mentioned -> aforementioned
- the tables/images are laid out awkwardly on the page, with labels being too close to other visuals or going over to another page.
- section 2 can be named more specifically, such as "Background and data for power monitoring" (or something similar)
- what is the "actual threshold of power consumption" in section 3? Needs to be clarified.
- "pieces of points" -> "groups of points" or "sets of points"
- in Tab 3, each column is presumably one cluster. If so, the first row shouldn't say "the number of clusters" but "the ID of the cluster" or "the number of the cluster"
- section 7 needs a forward reference regarding why K=3 for KNN throughout the paper. A short explanation needs to be given up-front.
- Figure 11 should have the colors described either on the plot itself or in the caption.
- it's not entirely clear why the term "eigenvalues" is used -- if it's in a literal sense with respect to some matrix/operator, then what is the matrix/operator? If not, then please explain and give a reference to the alternative meaning of "eigenvalues".
- instead of writing "slow" and "fast" for computing speed, why not give the exact numbers/ranges and let the readers judge for themselves what to consider slow and fast?
- what is the "eligible" identification accuracy?
Reviewer 4 Report
Research of public building energy efficiency is very important today.
In the paper proposes an online separation methodology for separating the power consumption of lighting sockets and air conditioning in public buildings.
Abstract - It does not clearly present the issues addressed in the paper.
Where is the discussion? Authors should discuss the results and how they can be interpreted in perspective of previous studies and of the working hypotheses. The findings and their implications should be discussed in the broadest context possible.
Round 2
Reviewer 1 Report
Thank you for addressing my concerns. I appreciate that authors replied my comments in detail.
Author Response
尊敬的编辑和审稿人:
感谢您对我们的稿件提出的意见和帮助。
最好的祝福,
马良东
Reviewer 2 Report
The revision and responses addressed all the raised questions. The manuscript deserves to be published.
Author Response
Dear Editor and Reviewers:
Thank you for your comments and kind work concerning our manuscript.
Best Regards,
Liangdong Ma
Reviewer 3 Report
Thank you to the authors for carefully updating the manuscript and adding the details and explanations. Now it is substantially easier to follow, and the technical contribution is well-explained. And the use of the language service is appreciated. So I can now withdraw my complaints about the presentation quality.
Just one more grammar rule to note: no matter how long and complex the subject it, this alone cannot justify putting a comma between the subject and the verb (only put it if there is another justification, such as a subordinate clause). Examples below.
Minor edits (indicating that the authors should proofread more):
- It's-> it's
- buildings ,it -> buildings, it
- Anshan (Building C), are -> Anshan (Building C) are
- air conditioning opened, are -> air conditioning opened are
- one is for abnormal daily data, the other is for abnormal daily data -> one is for abnormal daily data, the other is for abnormal hourly data
- air conditioning three kinds of state -> three kinds of air conditioning state
- data three kinds of state -> three kinds of data state
